# The Design of Rapid Self-Healing Alginate Hydrogel with Dendritic Crosslinking Network

**DOI:** 10.3390/molecules27217367

**Published:** 2022-10-29

**Authors:** Dingxuan Wang, Yuhan Li, Haobo Zhang, Zhaorong Ren, Kefan Fan, Jue Cheng, Junying Zhang, Feng Gao

**Affiliations:** School of Material Science and Engineering, Beijing University of Chemistry Technology, Beijing 100029, China

**Keywords:** alginate hydrogel, PAMAM, dendritic crosslinking network, rapid self-healing

## Abstract

Self-healing alginate hydrogels play important roles in the biological field due to their biocompatibility and ability to recover after cracking. One of the primary targets for researchers in this field is to increase the self-healing speed. Sodium alginate was oxidized, generating aldehyde groups on the chains, which were then crosslinked by poly(amino) amine (PAMAM) via Schiff base reaction. The dendritic structure was introduced to the alginate hydrogel in this work, which was supposed to promote intermolecular interactions and accelerate the self-healing process. Results showed that the hydrogel (ADA-PAMAM) formed a gel within 2.5 min with stable rheological properties. Within 25 min, the hydrogel recovered under room temperature. Furthermore, the aldehyde degree of alginate dialdehyde with a different oxidation degree was characterized through gel permeation chromatograph aligned with multi-angle laser light scattering and ultraviolet (UV) absorption. The chemical structure of the hydrogel was characterized through Fourier transform infrared spectroscopy and UV-vis spectra. The SEM and laser scanning confocal microscope (CLSM) presented the antibiotic ability of ADA-PAMAM against both *S. aureus* and *E. coli* when incubated with 10^−7^ CFU microorganism under room temperature for 2 h. This work presented a strategy to promote the self-healing of hydrogel through forming a dendritic dynamic crosslinking network.

## 1. Introduction

Alginate, a linear polysaccharide with β-D mannuronic acid (M) units and α-L guluronic acid (G) units, has already proved to be an excellent and economical backbone polymer for biocompatible hydrogels [1]. Most of the alginate hydrogels are prepared via non-reversible crosslinking strategies. For instance, calcium ions provide strong coordinated bonds accompanied by carboxyl groups on alginate, forming an ‘eggshell’ structure and inducing gelation [2]. On the other hand, the carboxyl groups are able to act as the crosslinking sites by forming amide bonds with amino-ended crosslinkers catalyzed by N-hydroxysuccinimide (NHS) and N-(3-dimethylaminopropyl)-N’-ethylcarbodiimide hydrochloride (EDC) [3]. Alginate hydrogel has been widely used in biological applications such as drug delivery, tissue engineering and artificial biological devices due to its soft structure, permeability and biocompatibility [4,5,6,7,8]. However, the crosslinking structure of the alginate hydrogel prepared via non-reversible bonding is permanently damaged after the physical shock, which limits its applications. Thus, the self-healing ability of hydrogel has caught many researchers’ attention in recent years [9].

The introduction of chemical or physical reversible interactions into the crosslinking network is one of the most straightforward ways of endowing the hydrogel with self-healing ability. Dynamic bonds recover after a period of time and the crosslinking networks rebuild. [10]. The Schiff base reaction forms the imine bond between the aldehyde group and amino group, producing a reversible covalent bond with excellent biocompatibility [11,12,13,14]. Rapid self-healing hydrogel provides a simultaneous repair when damaged that further prolongs its usage period. Schiff base self-healing hydrogel crosslinked by linear crosslinkers such as glutaraldehyde took over 12 h to recover, which was not able to repair the damaged network instantaneously. [15]. The increase of the imine bond concentration was able to accelerate self-healing [12,13,14]. Ding F. et al. [13] oxidized alginate by 50%, which was crosslinked by acrylamide-modified chitosan, and the fabricated hydrogel recovered in 1 h after the damage. Further increasing the imine concentration might affect the mechanical performance of the hydrogel and cause toxic issues. On the other hand, the optimization of the crosslinker structure was also able to boost the self-healing by enhancing the intermolecular interactions. Qu J. et al. [16] used a Pluronic F127-CHO micelle crosslink quaternized chitosan-forming hydrogel, and the self-healing process took around 2 h, which was supposed to be enhanced by the multiple functional crosslinking structure.

Poly (amidoamine) (PAMAM) is one of the most commercialized and well-studied dendrimers [17]. The dendritic branches of PAMAM facilitate the intermolecular interactions between the terminal functional groups and the neighbor molecules. As a result, PAMAM is widely used as the crosslinker and the vehicle in drug delivery and gene reparation [18]. PAMAM holds a well-defined structure, and the diameter is able to be controlled accurately by varying the generation. For example, 5th generation PAMAM (G5 PAMAM) possesses a spherical shape with a diameter about 5 nm [19]. Moreover, PAMAM has been used as the grafting arm for the surface modification of micro-fluid devices where the morphology was restricted. The polycationic nature endowed PAMAM with biocidal ability, which also caused the toxic issue [20]. However, according to our previous work, the cytotoxicity of PAMAM was greatly eliminated when the primary amine groups were shielded with the microbicidal ability preserved [21]. Furthermore, PAMAM can also be used to fabricate crosslinking networks, as recent studies pointed out. I. Matai et al. [22] used G5 PAMAM to crosslink alginate chains with the help of NHS and EDC through forming amide bonds. However, the permanent amide-bond network induces a less durable structure. Rumwald Leo G. Lecaros et al. [23] prepared the alginate membrane crosslinked by G0–G2 PAMAM. It was proposed that there was a strong interaction between sodium alginate and PAMAM which stabilized the sodium alginate chain. Moreover, it is clear that the more reaction sites a crosslinker molecule possesses, the more stable the interaction is [24]. Therefore, the abundant primary amide groups on one PAMAM molecule are proposed to promote the crosslinking process of PAMAM-crosslinked alginate hydrogel and thus, PAMAM is supposed to accelerate recovery.

Herein, PAMAM was introduced into the oxidized alginate, forming the reversible imine bonds and constructing the hydrogel with rapid self-healing ability. It was hypothesized that (1) the dendritic crosslinker could facilitate the intramolecular interactions, which would accelerate the exchange between the imine bonds, and endow the hydrogel rapid self-healing ability; (2) the mechanical properties of the hydrogel were tunable by varying the concentration of PAMAM and the oxidation degree (OD) of alginate; and (3) the polycationic nature of PAMAM could endow the hydrogel with biocidal ability against both Gram-positive and Gram-negative bacteria. Alginate oxidized by sodium periodate (alginate dialdehyde, ADA) was crosslinked with G5 PAMAM through Schiff base reaction, which was abbreviated as ADA-PAMAM (Figure 1). The molecular weight of alginate was reduced after the Malaprade reaction and influenced the mechanical performance of the hydrogel. Thus, the molecular weight before and after the oxidation was characterized via gel permeation chromatograph (GPC). The alginate degree (AD) of alginate was evaluated by the calculation of the aldehyde concentration on the chain, which was characterized with the help of aniline. Aniline was able to react with aldehyde, forming the complex with the maximum ultraviolet (UV) absorbance at 335 nm, and thus the aldehyde amount was able to be quantitatively characterized by the evaluation of the absorption strength at 335 nm with the help of UV extinction of the aldehyde-aniline complex. The morphology of hydrogel was confirmed by scanning electron microscopy (SEM), and the imine bond was characterized by Fourier transform infrared spectroscopy (FTIR) and UV-vis spectra. Rheological properties of hydrogel, including gelation time and modulus, were also examined. Self-healing ability at room temperature was estimated by the cyclic stress test and the crack recovery time, with the help of a rheometer and optical microscopy. The bactericidal ability of the hydrogel against *S. aureus* and *E. coli*. was evaluated by accounting the surviving bacteria after the incubation with the hydrogel ex vivo. The fabrication of ADA-PAMAM provides a strategy to introduce dendritic dynamic network into polysaccharide hydrogels to boost the self-healing.

## 2. Results and Discussion

### 2.1. Preparation of Alginate Dialdehyde with Different Oxidation Degree

Alginate was first oxidized by sodium periodate via Malaprade reaction in a mixture of ethanol and deionized (DI) water, generating alginate dialdehyde (ADA) as presented in Figure 1. A pair of aldehyde groups were generated on each repeating unit of ADA after the oxidation, and the oxidation degree (OD) was defined as the molar ratio between sodium periodate and the repeating unit of alginate. The ODs were chosen as 8%, 20% and 30% in this work. The aldehyde degree (AD) was defined as the average molar concentration of aldehyde groups on the alginate chains, which was characterized by gel permeation chromatograph aligned with multi-angle laser light scattering and UV-absorption (GPC-MALLS-UV).

GPC provided the Refractive Index (RI) signal with the elusion volume, and the weight average molecular weight was obtained by the linear fitting of the standards (G1–G5 PAMAM). The mass of the detected sample was calculated by the area of RI peak and the dn/dc of PAMAM. MALLS provided the dynamic multiple angle light scattering signal, and the molecular weight was calculated with the Zimm plot. Aniline reacted with the aldehyde from ADA, forming the complex which showed the maximum UV absorbance of around 335 nm, as presented by Figure 2. The aldehyde groups from the polysaccharide were quantitatively characterized via the UV absorption signal and the UV extinction coefficient of the aniline-aldehyde complex. The spectrums obtained from GPC-MALLS-UV for the alginate with different ODs after the aniline dyeing were also plotted in Figure 2. The solid lines in Figure 2 represent the RI signal. The RI signal of the original alginate shifted from left to right when the OD was 8%, 20% and 30%, respectively, indicating the decrease of the molecular weight. The Mw fitted from the standards of the alginate was 121 ± 15.2, and became 67 ± 6.3 kDa when oxidized by 8% sodium periodate, which was listed in Appendix A. The Mw kept decreasing with the increase of OD, which became 43 ± 4.1 kDa and 39 ± 3.7 kDa when the OD was 20% and 30%, respectively. The dash line in Figure 2 represents the UV absorption at 335 nm for all the samples. No signal was captured for the un-oxidized alginate, indicating no aldehyde groups were detected on the alginate. The UV absorption signal shifted to the right when the OD was 20%, and the absorption at 335 nm raised significantly. The AD for 8% oxidized alginate was calculated as 7.0 ± 0.9 (%) according to the UV absorption peak area and the UV extinction coefficient of the aniline-aldehyde complex. The *AD* kept increasing to 16.3 ± 2.2 (%) and 25.2 ± 2.5 (%) when the OD raised to 20% and 30%, respectively. The GPC-MALLS-UV characterization indicated that the aldehyde groups were generated on the alginate chains after the oxidation, and that the AD and molecular weight were tunable by varying the OD in this work.

### 2.2. Synthesis and Characterization of ADA-PAMAM through Schiff Base Reaction

PAMAM was introduced into ADA acting as the crosslinker. The terminal primary amine from PAMAM reacted with the aldehyde from the ADA chains via Schiff base reaction, forming the hydrogel as demonstrated by Figure 1. According to the OD of ADA, PAMAM hydrogels were divided into three groups, noted as ADA8-PAMAM, ADA20-PAMAM and ADA30-PAMAM, respectively. Figure 3a shows the solution of alginate (16 wt%) and PAMAM (36 wt%) in 0.01 M PBS buffer saline with a weight ratio of 1.2:1 (molar ratio between amine and polysaccharide monomer was about 1.85:1). Figure 3b presents 16 wt% ADA in DI water, and the white solution remained fluid, indicating that ADA was not able to crosslink by itself. The solution containing 16 wt% ADA (OD was 8%) and 36 wt% G5 PAMAM is presented in Figure 3c. The mixture was pale yellow, indicating the introduction of the Schiff base imine bond [25]. The mixture became non-flowable after 2 min, which remained in the top of the centrifuge tube, indicating the rapid gelation process. The solution of ADA (16 wt%) and G5 PAMAM (36 wt%) is presented in Figure 3d. The color of ADA30-PAMAM became deeper than that of ADA8-PAMAM, indicating the further increasing concentration of the imine bond in the mixture. The crosslinked hydrogel was dried by the lyophilization, and the network morphology of the cross section was observed by SEM, which is presented in Figure 3e (the cross sections of all three hydrogels are shown in the Appendix A). The crosslinking of the hydrogel was further characterized by FTIR, and the spectrums for alginate, ADA and ADA-PAMAM are plotted in Figure 3d. The carbonyl group in the alginate itself had an absorption peak in the range of 1800–1500 cm^−1^, and thus the absorption peak of the aldehyde group (C=O at 1732 cm^−1^) was overlapping. The peak located around 727 cm^−1^ referred to aldehyde groups [26], which was not significant in alginate. However, this peak increased significantly when oxidized by sodium periodate, indicating the formation of aldehyde groups, which was consistent with the GPC-MALLS-UV characterization results. The peak located around 727 cm^−1^ disappeared after the ADA reacted with PAMAM, indicating that the aldehyde groups were consumed by the Schiff base reaction. The imine bond had a stretching vibration peak around 1630 cm^−1^, which was also overlapped by the absorption peak of carbonyl group in alginate, 1630 cm^−1^. The absorption peak at 1456 cm^−1^ in ADA-PAMAM was supposed to refer to the C-N bond of the primary amide. The crosslinking of ADA-PAMAM was further characterized by UV-vis spectra, which is plotted in Figure 3f. A total of 0.5 mM zinc chloride was introduced to each sample, amplifying the Schiff base UV absorption peak [25]. The zinc chloride solution presented no significant UV absorption around 300 nm, indicating the small influence of zinc ions in characterizing Schiff base imide bonds. However, the absorption peak increased significantly when the OD increased to 8%, indicating the existence of the Schiff base reaction between PAMAM and ADA. The UV absorption around 300 nm kept increasing with the OD, which was 1.854 and 2.699 when the OD was 20% and 30%, respectively. The ratio of the absorption strength around 300 nm between ADA8-PAMAM, ADA20-PAMAM and ADA30-PAMAM was 3:2.06:0.81, similar to the AD ratio characterized by GPC-MALLS-UV which was 3:1.94:0.83.

### 2.3. Rheological Properties of ADA-PAMAM

The rheological behavior of the ADA-PAMAM was examined in real time, as plotted by Figure 4. The solid lines in Figure 4a indicate the storage moduli (G′) and the dash lines stand for the loss moduli (G″). The mixing between ADA and PAMAM took around 2.5 min before the starting of the test. As demonstrated by Figure 4a, the G′ was larger than G″ for all the samples. The corresponding loss coefficient for ADA8-PAMAM, ADA20-PAMAM and ADA30-PAMAM are, respectively, calculated and presented by Figure 4b, which was lower than one throughout the testing, indicating that the gelation time for all the formulations was lower than 2.5 min. The gelation was evaluated as completed when the storage modulus was steady. The storage modulus of ADA8-PAMAM was about 247 Pa, which was around one order lower than that of ADA20-PAMAM and ADA30-PAMAM, indicating the positive relationship between AD and the storage modulus. However, the storage modulus of ADA20-PAMAM was around 1336 Pa, which was higher than that of ADA30-PAMAM, which was about 707 Pa. This was proposed to be related to the synergistic effect between the steric hindrance of PAMAM and the average backbone length of ADA. Due to the fixed size, steric hindrance of PAMAM prevented further contact between amino groups and aldehyde groups when crosslinking shorter ADA chains as the AD increased, which also hindered the speed of self-healing, as shown in Figure 4b.

Figure 4c shows the frequency sweep of the rheology evaluation. Both G′ and G″ increased with the frequency, however, both kept at a constant order of magnitude. G′ was significantly higher than G″ for all the samples with a frequency ranging from 0.01–100 rad/s, indicating the crosslinking structure was not damaged during this process. The G′ of all the samples kept increasing during the test, indicating that the Schiff base reaction kept occurring after the rapid gelation. It was indicated by the GPC result that the molecular weight was decreasing with the increasing of the OD, however, the G′ of ADA30-PAMAM remained the highest among these three samples during the whole testing process, which indicated that the increase of the aldehyde concentration on alginate greatly enhanced the crosslinking degree. The amplitude sweep of the rheology evaluation of the hydrogel with different formulation was presented by Figure 4d. G′ and G″ for these three samples kept steady when the shear strain was below 100%, but fluctuated wildly when the shear strain further increased, which indicated the crosslinking network for these three hydrogels began to collapse when the shear stain exceeded 100%. When the shear strain reached 350%, the loss modulus of all three hydrogels were higher than the corresponding storage modulus, indicating a completely collapsed hydrogel network [27].

### 2.4. Self-Healing Ability of ADA-PAMAM

The self-healing ability of ADA-PAMAM was evaluated, as presented by Figure 5 and Figure 6. ADA20-PAMAM presented as orange and was cut into two pieces, which were then contacted together under room temperature. The separated pieces were connected after 25 min as presented by Figure 5a–d. ADA8-PAMAM and ADA20-PAMAM were then contacted together, and became connected after 25 min healing. Results indicated ADA-PAMAM was able to reconstruct the crosslinking network due to the reversible imine bonds even if the formulation was different. The morphology of the connection crack between two pieces of ADA20-PAMAM was monitored by optical microscopy, as presented by Figure 5g–j. The crack became narrow after 5 min under room temperature, and illegible when the healing time was 15 min. The hydrogel was totally self-healed after 25 min, where the cracks were totally eliminated. According to the previous work [12], the crack in self-healing Schiff base alginate hydrogels with linear crosslinkers required about one hour to eliminate, while the crack in ADA-PAMAM disappeared within 25 min. This indicated the dendritic crosslinking networks possessed an accelerated self-healing process compared to linear crosslinking networks, which was consistent with the raised hypothesis that the dendritic crosslinking structure facilitated the intermolecular interaction, which promoted the exchange reaction between crosslinking segments, increasing the self-healing speed.

The self-healing ability was further evaluated by the cyclic stress testing with the help of a rheometer, as presented by Figure 6. The frequency was set as 1 Hz, and the amplitude was set as 1% for the first 600 s, since the hydrogel crosslinking network kept steady under this condition according to the results of the frequency sweep and amplitude sweep. The shear stain was increased to 350% when the testing time reached 600 s, under which condition the hydrogel crosslinking network was damaged. As demonstrated by Figure 6b, the G′ of ADA8-PAMAM was around 90 in the first 10 min and dropped immediately to the same level as the G″ when the shear strain increased to 350% due to the collapse of the crosslinking network. Both the G′ and G″ kept fluctuating around 10 Pa until the shear strain turned back to 1% at 1200 s, after which the G′ recovered from 10 Pa to 72 Pa (80% as the initial value) gradually within 30 s. This result was consistent with the results of the optical microscopy that the crosslinking network was able to recover after collapse due to the reversible exchange reaction of the imine bonds between the hydrogel fragments. ADA8-PAMAM was able to recover after the second cycle, and the change of the moduli showed no significant difference with that of the first cycle. Similar trends are presented in Figure 6c,d, where the AD increased to 20% and 30%, respectively. The initial G′ for ADA20-PAMAM and ADA30-PAMAM were 550 Pa and 1089 Pa, respectively, which was consistent with the rheology testing. The self-healing process was significantly faster for the hydrogel with higher AD. G′ recovered to its initial value almost instantaneously after the restoration of the shear strain for ADA20-PAMAM and ADA30-PAMAM. The reason was that the higher concentration of aldehyde groups greatly promoted the exchange reaction between the imine bonds. Interestingly, the G′ for ADA20-PAMAM and ADA30-PAMAM was even higher than the initial values after the self-healing. It was speculated that the crosslinking network was optimized during the healing process, and the evaporation of water may also increase the moduli.

### 2.5. Antibiotic Ability of ADA-PAMAM

The biocidal ability of the hydrogel was examined by accounting for the surviving bacteria after incubation with the hydrogel, and the testing process was demonstrated by Figure 2. A total of 1 mL of *S. aureus* and *E. coli* with 10^-7^ CFU was incubated with 0.2 g ADA-PAMAM for 2 h under 37 °C in a CO_2_ incubator with a piece of polyethylene glycol terephthalate (PET) slide. A total of 1 mL of the solution was titrated in the well of a 12-well plate. The morphology of the surviving bacteria on the PET slides is presented in Figure 7a. The PET slide was covered by spherical *S. aureus* without the participation of the hydrogel. However, the number of bacteria reduced significantly when incubated with ADA-PAMAM after 2 h (Figure 2). At the same time, blebbing was presented by *S. aureus* after incubation, indicating the lysis of the bacteria [28], which was the typical behavior of *S. aureus* when in contact with PAMAM due to the electron static interaction between the terminal amine and the cell membrane. The increasing of the AD for the hydrogel did not significantly influence the surviving *S. aureus* concentration as presented by Figure 7b–d, and blebbing was also observed in these samples. The live/death accounting result was consistent with the SEM images, showing that the number of living *S. aureus* reduced significantly with the introduction of ADA8-PAMAM, and kept at the same level when the OD increased to 20% and 30%. This probably resulted from the lower concentration of free amino groups in ADA8-PAMAM, as the previous study mentioned [29]. Figure 7e–h presents the morphology of *E. coli* on the PET slide after culturing. The total amount of living *E. coli* was significantly lower compared with that of *S. aureus*, as *E. coli* survived without attaching to the substrate. A similar trend was presented by *E. coli*; the number of living *E. coli* reduced significantly after the introduction of ADA-PAMAM, and showed no significant change when the OD increased from 8% to 20% and 30%. The live/death laser scanning confocal microscope (CSLM) images presented by Figure 7 were consistent with the SEM characterization, in which the red dots refer to dead microbials and green dots refer to living ones, respectively. The results above indicated the introduction of PAMAM endowed the ADA-PAMAM with biocidal ability, and the hydrogel after crosslinking was able to kill both Gram-positive and Gram-negative bacteria in the solution.

## 3. Materials and Methods

### 3.1. Materials

Sodium alginate (the ratio of M structure and G structure was 2:1 [30]) was purchased from Macklin, Shanghai, China. Generation five poly (amidoamine) dendrimer (G5 PAMAM, 20 wt% methanol solution) was purchased from the Chenyuan Company, Weihai, China. Sodium periodate was also obtained from Macklin. PBS buffer powder was purchased from Biosharp, Hefei, China. Staphylococcus aureus (*S. aureus*) and Escherichia coli (*E. coli*.) were obtained from Puboxin Bio-Technology Co., Ltd., Beijing, China.

### 3.2. Preparation of Alginate Dialdehyde (ADA)

Sodium alginate was oxidated by sodium periodate through a water-ethanol solution to guarantee enough yield of ADA [31]. First, 5 g sodium alginate was mixed with 25 mL ethanol to prepare the sodium alginate ethanol suspension. To ensure the uniformity, the suspension was stirred through a magnetic stirrer. Sodium periodate was dissolved in deionized (DI) water and poured into the suspension. Due to the molecular weight of sodium periodate being similar to the average weight of sodium alginate, three groups noted 8% oxidized, 20% oxidized and 30% oxidized were divided according to the weight ratio of sodium periodate and that of sodium alginate which was defined as the oxidation degree (OD). The reaction lasted at least 6 h and was stopped by adding overdosing ethylene glycols. A total of 250 mL ethanol was added into the mixture and deposited overnight, then the supernatant was removed. The remaining 125 mL ethanol was added and stirred uniformly, and the mixture was centrifuged and the supernatant was then removed. This process was repeated 2 times. Finally, the mixture was frozen under -50 °C for 12 h and then lyophilized. The lyophilized powder was stored evading vapor under 4 °C. Aldehyde degree was confirmed through gel permeation chromatography (GPC, Waters e2695, MA. Massachusetts, America) aligned with multi-angle laser light scattering and UV absorption (Waters 2489) (GPC-MALLS-UV) with the help of aniline [32]. Lyophilized powder was dissolved by DI water at the concentration of 25 mg/mL, while aniline acetate reagent (10%, *v*/*v*) was added into the solution (90 μL) with a volume ratio of 1:9. The specimen solution was incubated for one hour at room temperature and then diluted to 0.5 mL.

### 3.3. Preparation of PAMAM Crosslinked Alginate Hydrogel (ADA-PAMAM)

PAMAM was dried to remove methanol. Hydrogels were prepared by straightly mixing 36 wt% PAMAM aqueous solution with 16 wt% DI water solution, noted as ADA-PAMAM. According to the OD of ADA, three groups, which were noted ADA8-PAMAM, ADA20-PAMAM and ADA30-PAMAM, were divided, corresponding to the 8% oxidized, 20% oxidized and 30% oxidized ADA, respectively. The weight ratio of the ADA and PAMAM was 1.2:1, which was controlled at 10% through PBS buffer saline. Each group hydrogel was prepared with the help of rheometer. PAMAM was reduced first and then ADA was reduced. The gelation occurred between a gap of 0.5 mm and lasted 100 min. After this, hydrogel was carefully transferred from a rheometer to a 1 mL centrifuge tube and stored at 4 °C.

### 3.4. Characterization

Bruker NanoIR (Bruker, Germany) was used to perform the Schiff base structure in ADA-PAMAM. Since there were serious interference peaks when characterizing hydrogel in the wet state, we lyophilized the hydrogel beforehand. Schiff base structure was further confirmed through UV-vis spectra (UV-2550, Shimadzu, Beijing, China). Hydrogel was diluted to a concentration of 1 mM. To enhance the absorption peak of the Schiff base, zinc ions were introduced as a concentration of 0.5 mM. Scanning electron microscopy (SEM, ZEISS GeminiSEM 300, Oberkochen, Germany) was used to observe morphology in cross sections of hydrogels with gold sputtering. With the help of a mold, hydrogels were formed into bars (shown in Appendix A). Each hydrogel bar was lyophilized for 48 h. Lyophilized hydrogel was cut by a surgical blade.

### 3.5. Rheology Test

A rheology test was performed through a modular compact rheometer (Anton Paar, Graz, Austria). Considering that hydrogel was also prepared by a rheometer, a time sweep test was performed simultaneously with a gap of 0.5 mm. Both a frequency sweep and amplitude sweep of a newly made hydrogel were performed with a gap of 1 mm. For the frequency sweep test, the shear strain was set at 1%, while the frequency was set at 1 Hz for the amplitude sweep test. The hydrogel was ensured to be wet throughout the test.

### 3.6. Self-Healing Characterization

ADA20-PAMAM was cut into two pieces and combined to examine the self-healing ability. ADA8-PAMAM and ADA20-PAMAM were also combined to confirm the connection between ADA-PAMAM with different formulations. ADA20-PAMAM was observed through an ultra-depth-of-field optical microscope (Leica DM4-6, Märzhäuser Wetzlar GmbH, Wetzlar, Germany) at room temperature. The magnification of the ultra-depth-of-field OM was 377×. A surgical blade was used to make a crack on the surface of ADA20-PAMAM, and then the hydrogel was put onto the platform with the crack facing up. Self-healing ability was further performed through cyclic stress test was performed on the rheometer with a gap of 1 mm. Each group of ADA-PAMAM was preserved under 4 °C within 2 h to reduce the impact of degradation. Herein, we alternated the shear strain between 1% and 350%, since ADA-PAMAM showed a liquid-like state over a 350% shear strain while a gel-like state was observed under 1% shear strain. Each phase of the cyclic stress test lasted 10 min and shear strain was alternated for 5 times.

### 3.7. Bactericidal Assay

*S. aureus* and *E. coli* were used for the antimicrobial assays of ADA-PAMAM. Each bacterium was diluted in a concentration of 10^−7^ CFU/mL using 0.01 M PBS buffer saline. Considering that it is difficult for bacteria to immobilize on the hydrogel, we observed the surface of a polyethylene glycol terephthalate (PET) film placed adjacent to the hydrogel instead, as presented in Figure 1. The PET film (Xingqiao company, Shanghai, China) was cut into square pieces with a length of 10 mm. All PET segments were cleaned in DI water and ethanol, respectively, by ultrasonic (Light Co., Ltd., Shenzhen, China) for 10 min, which were then dried under vacuum in 37 °C overnight. Each group of hydrogels and a piece of PET film was placed in the well of a 12-well plate, as well as the comparative group. A total of 1 mL of two kinds of bacterial nutrient solution was added to each well in a total of 8 groups. The film was ensured to be immersed in a bacterial nutrient solution. The 12-well plate with the specimen was incubated at 37 °C for 2 h and then the culture solution was aspirated using a syringe. A total of 1 mL glutaraldehyde (GA) 3% aqueous solution (prepared by GA from Mairuier, Shanghai, China) was dropped into each well and the 12-well plate was stored overnight at 4 °C to immobilize bacteria on the surface of the PET. After removing the GA solution, the specimen was dehydrated with 20%, 40%, 60%, 80% and 100% ethanol aqueous solution and then stored in dry environment. The bacteria on the surface of the PET film were observed by SEM (JEOL, JSM-6700M) at an accelerating voltage of 5 kV with the same magnification for each group. The live/death situation was performed by a laser scanning confocal microscope (CLSM). A total of 1 mL of solutions A and B of AAT-22411 live/death kit was added into 100 μL PBS buffer saline, which was then titrated into the well in the 12-well plate. The dye process was performed under room temperature and lasted one hour. The filter FITC/TRITC was used, and 488 nm and 540 nm were chosen, which were observed as green and red, respectively.

## 4. Conclusions

An alginate hydrogel with tunable mechanical performance, rapid self-healing and biocidal abilities was designed and prepared in this work. Sodium alginate was oxidized by sodium periodate via Malaprade reaction, generating aldehyde groups on the chains. The molecular weight and the AD were quantitatively characterized via GPC-MALLS-UV evaluation with aniline dying, and were able to be controlled by tuning the OD, which further influenced the storage moduli positively. PAMAM, containing primary amine groups on the terminal, acted as the dendritic crosslinker in this work. The abundant imine bonds endowed the hydrogel with self-healing ability, and the separated segments were able to covalently connect within 25 min with all the cracks eliminated. The dendritic crosslinking structure increased the intermolecular interactions, which greatly reduced the gelation time and accelerated the self-healing. The rheology tests indicated the gelation time for all the formulations was within 2.5 min, and the collapsed crosslinking network recovered almost instantaneously when the shear strain reduced from 350% to 1%. The prepared hydrogel was endowed with biocidal ability due to the polycationic nature of PAMAM. SEM results indicated that ADA-PAMAM was able to kill both Gram-positive and Gram-negative bacteria when incubated together in the solution under room temperature. The rapid self-healing ability, the ease of preparation, the biocompatible character, the tunable mechanical properties and biocidal ability made ADA-PAMAM a bright future for clinical applications. Moreover, the abundant primary amine endowed ADA-PAMAM with the ability to be further functionalized.

## Data Availability

Not applicable.

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
