# Peer review of "The Design of Rapid Self-Healing Alginate Hydrogel with Dendritic Crosslinking Network"

_molecules, 2022, doi:10.3390/molecules27217367_

Round 1

Reviewer 1 Report

I think it is a good manuscript. The organization of the manuscript is reasonable. I recommend that it for publication. The authors prepared ADA-PAMAM hydrogel with rapid self-healing ability and antibiotic ability by Schiff base reaction between oxidized alginate and PAMAM. The effects of the oxidation degree (OD) of ADA on the rheological behavior, self-healing ability and antibiotic ability of the ADA-PAMAM hydrogel were investigated. The results essentially support the author's claims, and the organization of the manuscript is reasonable. I recommend it publication.

Author Response

Thank you very much for your attention and recommendation of our article to The Design of Rapid Self-Healing Alginate Hydrogel with Dendritic Crosslinking Network. We do appreciate the constructive and quick feedback and we glad to share our recent work.

Reviewer 2 Report

The manuscript molecules-1928631 "The Design of Rapid Self-Healing Alginate Hydrogel with Dendritic Crosslinking Network" by Wang et al. describes the design and synthesis of self-healing hydrogel based on alginate and 5th generation PAMAM dendrimer and the study of their physical and antimicrobial properties. The authors have interesting experimental results, so I think that this paper will be of interest to the readers of Molecules.

Questions and comments:

1) The manuscript contains many abbreviations. A list of abbreviations should be added.

2) Can the authors somehow quantify antimicrobial activity (MIC, MBC, etc.)?

3) I recommend the authors to compare the obtained results with those known in the literature.

4) Please recheck the upper and lower subscripts in the text (e.g. CO2, cm-1, 10-7).

5) Please sign which ADA-PAMAM is depicted on each insert in Figure 5.

6) What glycols did the authors use to stop the oxidation reaction (part 3.2)?

7) Minor changes:

- Line 51. "2 hours" is duplicated.

- Line 64 "a diameter of 5 μm". Maybe a diameter of 5 nm? Please add a reference to literature data.

- Line 85 "OD of alginate". An abbreviation is usually defined for the first time in the text.

- Bacterial strains should be written in italics.

- Line 157 What is "1× PBS"?

Round 2

Reviewer 2 Report

I thank the authors for answering my questions and improving the manuscript.